# Management of Lung Cancer Presenting with Solitary Bone Metastasis

**DOI:** 10.3390/medicina58101463

**Published:** 2022-10-16

**Authors:** Claudiu-Eduard Nistor, Adrian Ciuche, Anca Pati Cucu, Cornelia Nitipir, Cristina Slavu, Bogdan Serban, Adrian Cursaru, Bogdan Cretu, Catalin Cirstoiu

**Affiliations:** 1Department Thoracic Surgery, Dr. Carol Davila Central Military Emergency University Hospital, 010825 Bucharest, Romania; 2Department of Medical Oncology, Carol Davila University Medicine & Pharmacy, Elias University Emergency Hospital, 011468 Bucharest, Romania; 3Department Orthopedic & Traumatology, Carol Davila University Medicine & Pharmacy, University Emergency Hospital, 050098 Bucharest, Romania

**Keywords:** lung cancer, solitary bone metastasis, bone metastasis resection reconstruction, radical therapy, targeted therapy

## Abstract

Lung neoplasm is the main cause of cancer-related mortality, and bone metastasis is among the most common secondary tumors. The vast majority of patients also present with multiple bone metastases, which makes systemic and adjuvant pain therapy preferable to surgery. The optimal approach for a resectable non-small-cell lung tumor that also presents a unique, resectable bone metastasis is not fully established. The number of papers addressing this subject is small, and most are case reports; nevertheless, survival rates seem to increase with radical surgery. The sequencing of local versus systemic treatment should always be discussed within the multidisciplinary team that will choose the best approach for each patient. As targeted systemic therapies become more accessible, radical surgery, together with existing reconstructive methods, will lead to an increase in life expectancy and a better quality of life.

## 1. Introduction

Lung cancer is the leading cause of cancer-related deaths, and bone metastasis is one of the most prevalent secondary malignancies. Because the vast majority of patients have numerous bone metastases, systemic and adjuvant pain treatment is preferred over surgery. With an average survival of less than a year, these patients have a poor prognosis [1].

Bone metastases are associated with increased morbidity, loss of function, and decreased quality of life [2,3]. The main locations of metastases secondary to non-small-cell lung cancers (NSCLC) are the spine in 50% of cases, the ribs in 27.1%, the iliac bone in 10%, the sacrum in 7.1%, and the femur in 5.7% of cases, followed by the humerus, scapula, and sternum at 2.9% [4]. We noted from this study that, in most cases, the curative treatment for a bone metastasis secondary to a lung neoplasm could not consist of a resection due to the anatomical location.

Bone metastases are often asymptomatic and represent the first symptom in only 2.5% of lung neoplasm cases [5]. During the spread of the disease, 80% of patients with lung neoplasm complain of bone pain [6].

A very small number, approximately 7%, of patients diagnosed with non-small-cell lung neoplasm present a single metastasis at the initial evaluation [7]. According to current published guidelines, a non-small-cell lung neoplasm with a single brain or adrenal metastasis can be removed by synchronous metastasectomy if the lung tumor is also resectable [8,9,10].

The best course of action for a resectable lung tumor that also has a solitary, resectable bone metastasis is still up for debate. Despite the fact that there are few articles on this topic and that the majority of them are case reports, after extensive surgery, patient survival rates appear to improve [11,12].

The present study analyzed a small category of patients with lung neoplasms that were associated with single bone metastasis. Synchronous surgical treatments to resect the non-small-cell lung neoplasms lead to significant increases in medium- and long-term survival. We reviewed 41 papers that presented several unique bone metastases that were secondary to pulmonary malignancies.

## 2. Pathophysiology of Bone Metastasis

Bone neoplastic development results from an alteration in the perfectly balanced bone-remodeling cycle between bone resorption by osteoclasts and bone tissue production mediated by osteoblasts [13]. Metastatic pathophysiology consists of excessive osteoclastic stimulation that leads to bone resorption. The bone matrix releases cytokines that stimulate tumor growth, thus creating a vicious circle of bone resorption and tumor stimulation. The main culprits in this mechanism are parathyroid-hormone-related peptides (PTHrP) and Interleukin 8 [14,15,16]. Disruption to the receptor activator of the nuclear factor kappa (RANK)–receptor activator of the nuclear factor kappa B ligand (RANKL)–osteoprotegerin (OPG) axis is the basis for the development of bone metastasis [17,18]. Those that are secondary to lung neoplasms are osteolytic, but osteoblastic forms have also been described.

An important aspect of the development of osteolytic bone metastases is the secondary hypercalcemia that can occur. Correct hydration, diuretics, and even glucocorticoids can be used in these cases. The intravenous administration of bisphosphonates, together with adequate hydration, can lead to the normalization of calcium levels [19].

## 3. Skeletal Metastasis Secondary to Lung Cancer

Lung neoplasms are among the most common primary malignancies that metastasize to bones. Patients with a bone tumor lesion are divided into two categories: those with and without a known history of lung neoplasm. In the face of a suspicious bone lesion, a primary bone neoplasm must always be excluded. Within a multidisciplinary team, all decisions regarding staging and preoperative diagnosis are made before the treatment begins. A bone biopsy is mandatory to establish the origin of the primary tumor formation.

A separate category of patients is represented by those who present at the emergency room with a pathological bone fracture. Their staging and preoperative diagnosis must follow the same steps, although the fracture itself may represent an emergency.

The preoperative imaging evaluation is very important because it must make sure that the bone lesion is unique; only when this has been confirmed can curative treatment be discussed. In this context, a full imaging evaluation is required (thoracic–abdominal and pelvic-computed tomography (CT) scan, brain MRI, and bone scintigraphy or whole-body PET). Mediastinoscopy is becoming mandatory to exclude other secondary determinations.

The incidence of single bone metastases is difficult to determine, given the rarity of cases. The vast majority of published works are case reports or limited case series; nevertheless, a resectable lung neoplasm with a resectable single bone metastasis must be treated aggressively so that long-term survival can be improved.

A 2016 study that analyzed five non-small-cell lung cancer (NSCLC) patients exhibiting solitary bone metastasis concluded that the progression-free survival time improved on those rare occasions when synchronous aggressive surgical management was performed [20].

Another study, published in 2005, analyzed the evolution of two cases of lung neoplasm with single bone metastasis and found a survival rate of over five years for patients treated with surgical resection [12]. Another case report presented a patient who survived for more than eight years from diagnosis after a metastasectomy [11]. This approach may seem to have been a solution in these cases, but it should be performed only when the primary tumor can be controlled and R0 resection is possible [21,22].

A correct and complete evaluation by an experienced multidisciplinary team should guide the diagnosis and therapeutic strategy for this category of patient.

## 4. Sequencing of Treatment

Oligometastatic NSCLC refers to a limited number of metastases in a limited number of organs. The biology of the disease in these patients may have a slower evolution. The standard number of metastases has not been established, but a patient with fewer than five is considered oligometastatic. Their presentation may occur at the same time as the diagnosis or at a distance from definitive treatment [23].

The sequencing of local versus systemic treatment is not yet established. It is preferable to start with systemic treatments to verify the mild nature of the disease, but many clinical situations––the presence of a cerebral or bone metastasis with the risk of fracture or pain––require local treatment first [24].

The choice of systemic treatment is made according to the molecular profile of the tumor. At the metastatic NSCLC stage, broad molecular profiling of the tumor is indicated to identify all biomarkers in a single analysis and to include biomarkers being evaluated for targetability. However, treatment availability depends on the reimbursement condition and accessible clinical trials.

Among the more important biomarkers we mentioned are PD-L1, EGFR mutations, ALK translocation, K-RAS, RAS1, BRAF, NTRK1/2/3, RET rearrangement, and METex14. Broad molecular profiling should be considered for lung adenocarcinoma and squamous cell carcinomas alike [25].

If the molecular tests are negative for all of the above, systemic treatment is selected according to PD-L1. If the PD-L1 values are negative or below 50%, the alternatives are combined immunotherapy and chemotherapy; doublet immunotherapy; or combined immunotherapy, chemotherapy, and antiangiogenic treatment. If the PD-L1 values are over 50%, immunotherapy alone will be chosen [26].

Table 1 shows the systemic treatment options when targetable molecular alterations are present [27].

## 5. Indications for Surgical Treatment

### 5.1. Surgical Treatment Indications and Techniques Used in Single Bone Metastases Associated with Lung Cancer

There is no generally valid management option for patients presenting with lung cancer and single bone metastasis in the extrathoracic area (M1b metastasis staging site according to the TNM 8 classification of lung cancer). Therapeutic intervention is usually individualized according to prognostic factors (prognosis after bone metastasis, the number of metastases, performance status) and pain level [28,29,30].

The aim of the primary approach to the bone mass is to establish with certainty its metastatic origin and alleviate symptoms. Metastases at the level of the thoracic cage, in addition to being painful, interfere with respiration [31]. This may exacerbate lung-cancer-related respiratory impairment; on the other hand, respiratory dysfunctions affecting tolerance to general anesthesia can also make surgery questionable [30].

Surgical treatment of lung cancer can be performed simultaneously with the resection of the bone tumor within oncological limits if it is located ipsilaterally or at a distance, possibly after several courses of conversion chemotherapy, depending on the tumor location and bronchial tree status (Figure 1).

Because the primary lung tumor may progress during the recovery period that would be required after a bone metastasectomy, the preferred initial approach to the primary tumor is by various types of anatomic and nonanatomic lung resection and radical mediastinal lymphadenectomy.

Depending on the location, size, and extension of the lung tumor and respiratory and general functional status, several types of classical/open or minimally invasive surgery can be performed, the latter being preferred due to reduced pain and a shorter hospital stay. The following allow a faster recovery so that the patient may continue the therapeutic algorithm (orthopedic surgery versus adjuvant chemotherapy): nonanatomic (wedge) resections and anatomic resections (segmentectomies, lobectomies, bilobectomies) and pneumonectomies. Of these, lobectomy has been shown to increase overall survival rates in patients with NSCLC and a single metastasis compared to other types of lung resection [32].

Furthermore, it has been established that surgical intervention for bone metastases should take into account expected survival times [33].

Biopsies of single bone metastases are mandatory for diagnosis in order to exclude primary synchronous bone tumors, which require a different therapeutic approach [34].

Radical lung resection is only possible when the lymph node component is N0–N2, with N3 requiring neoadjuvant chemotherapy.

The resection of bone metastases may precede lung resection in several situations, such as pathological bone fractures [35]

In the presence of a significant chronic algic syndrome secondary to bone metastasis, the sequence of interventions consists of a resection (within oncological limits) of the bone tumor followed by surgical treatment of the lung cancer with or without neoadjuvant chemotherapy [36].

In the case of neurological damage (paresis, paralysis), spinal decompression surgery is performed first (M1b spine vertebral decompression), followed by lung surgery [34,35].

In addition, for spinal metastases with a survival expectancy of more than six months, open surgery is recommended, usually followed by radiotherapy [37].

Regarding resection expansions, the survival rate in the metastatic bone stage of lung cancer is so poor that an extended perimetastatic resection is not justified [38].

M1b-rib requires a resection within the oncological limits of the affected rib segment with parietal reconstruction for the prevention of paradoxical breathing.

M1b-sternum requires a partial resection of the sternum (manubrium +/− resection of the clavicular head or internal third of the clavicle +/− adjacent costal cartilages, or body, depending on location) followed by solid and soft tissue reconstruction. The reconstruction can be performed either with synthetic materials (polypropylene mesh, methylmethacrylate) or different cutaneous or myocutaneous flaps [39].

In patients with M1b-clavicle, a partial or total claviculectomy may be performed and is followed, or not, by reconstruction to restore the scapular girdle function [40,41].

M1b-scapula requires a partial or total resection associated with the resection of the head/external third of the clavicle/humeral head according to the invasion type.

There are circumstances in which bone metastasis has invaded the adjacent soft tissue, requiring extensive resection (thoracic parietectomies) and scapular resection along with the adjacent invaded musculature.

### 5.2. Resection and Reconstruction of Solitary Bone Lesions in Lung Cancer

Following an interdisciplinary consensus, the correct therapeutic procedure is indicated. A single metastasis secondary to a lung carcinoma will be treated by a wide surgical resection within oncological limits, followed by a reconstruction with modular prostheses (Figure 2). The resection followed by reconstruction is the most effective technique for maintaining postoperative function and preserving quality of life, given that secondary interventions will be avoided, such as those that occur following the degradation of osteosynthesis materials.

Currently, the vast majority of joints can be reconstructed; moreover, an entire femur or humerus can be prosthetically reconstructed if the following series of requirements are respected: correct preoperative planning, precise resection margins, correct restoration of the length of a limb, optimal fixation to bear important loads, and the restoration of muscle insertion, so that muscle function is not altered.

In particular, metastases located at the level of the femoral neck can be treated by arthroplasty using long femoral stems. Cephalomedullary rods are not an option due to the high mechanical stress from the neck and the lesion that remains, which bcomes progressively larger.

### 5.3. Treatment of Complete or Impending Fractures in Unique Bone Metastasis following Lung Cancer—Resection Followed by Intramedullary Nails and Acrylic Bone Cement

When facing a fractured or prefractured bone metastasis, even if it is unique, clinicians must balance whether a resection followed by reconstruction with a tumoral prosthesis respects the desired goal and, more precisely, if the resection should be R0. In these cases, it is preferable to perform as wide a resection as possible and use centromedullary rods reinforced with acrylic cement as a reconstructive method, followed by careful imaging and monitoring (Figure 3). During the second phase of treatment, the osteosynthesis method is replaced by a modular prosthesis. It is a method that allows waiting for the distal femur and proximal tibia not only until the oncological prognosis is good but until a definitive reconstruction can be performed.

## 6. Conclusions

Due to the relatively small number of patients with a lung neoplasm and single metastasis or oligometastasis, there are no large-scale randomized studies that investigate the therapeutic approach with certainty. Following the reporting of some case series and case presentations, it can be concluded that survival is considerably increased for these patients after radical therapy.

The sequencing of treatment will be conducted by discussing each clinical case within the multidisciplinary team. Depending on the location of the metastasis and its evolution, treatment can sometimes be urgent, such as a pathological bone fracture or pre fracture stage.

Correct and radical surgical treatment following a multidisciplinary consensus, together with the existing reconstructive methods, can lead both to an increase in life expectancy and an increase in the quality of life.

As targeted systemic therapies become more accessible, radical surgical treatment, along with local treatments, will increase the survival of these patients.

## Figures and Tables

**Figure 1 medicina-58-01463-f001:**
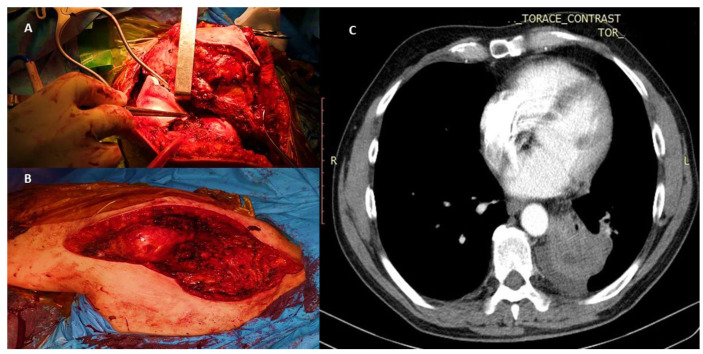
A 65-year-old man with a pulmonary neoplasm of the left-upper lobe confirmed by fibrobronchoscopy with right-scapular metastasis and invasion of adjacent soft tissues. Prior to radical pulmonary surgery, the patient underwent partial resection of the scapula (internal third), acromion, coracoid process, and lateral end of the clavicle (external third), together with the invaded soft tissue (subscapularis, suprascapularis, and infraspinatus muscles; humeral and clavicular insertion of the pectoralis major muscle; and insertion of the pectoralis minor and deltoid muscle). (**A**) Intra-operative view of scapular tumor mass (external third) invading the soft tissues and integument. (**B**) Post-excision tumoral aspect with soft tissue reconstruction that did not require prosthetic materials or skin graft. (**C**) CT scan of a left-upper lobe lung tumor diagnosed as lung cancer by fibrobronchoscopy.

**Figure 2 medicina-58-01463-f002:**
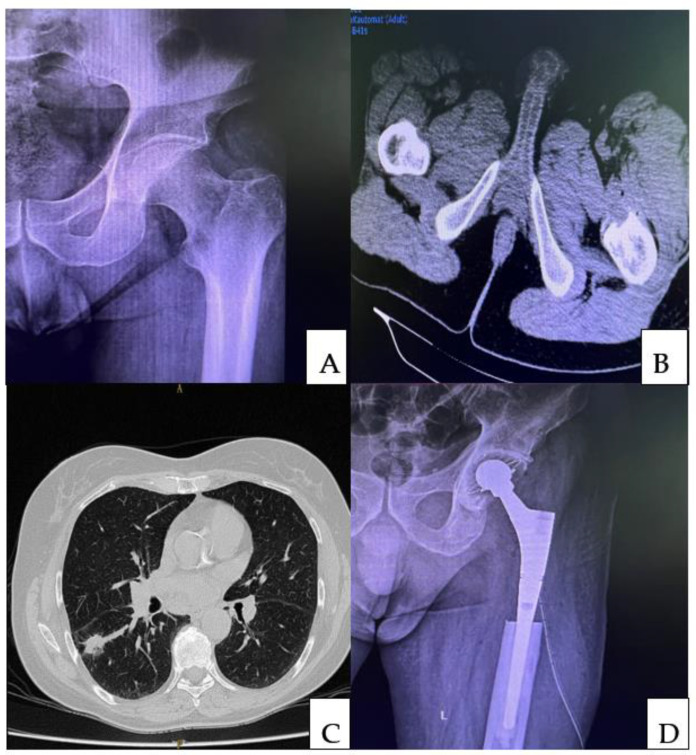
Pre-operative X-ray (**A**) of a 49-year-old male complaining of pain in the left hip that started three months earlier. An area of osteolysis at the level of the lesser trochanter was detected on the radiograph of the left hip. A computed tomographic (CT) scan of the pelvis (**B**) was performed, which revealed an osteolytic tumor at the level of the lesser trochanter. Following the biopsy, a diagnosis of bone metastasis secondary to a lung carcinoma was made. A chest CT scan revealed a spiculated, iodophilic nodule with retraction of the overlying pleura at the level of the apical segment of the right lower lobe (S6—Fowler) (**C**). The preoperative staging was a pulmonary tumor with single bone metastasis. Following multidisciplinary consensus, the patient was subjected to the radical resection of the pulmonary tumor followed by reconstruction of the left proximal femur (**D**).

**Figure 3 medicina-58-01463-f003:**
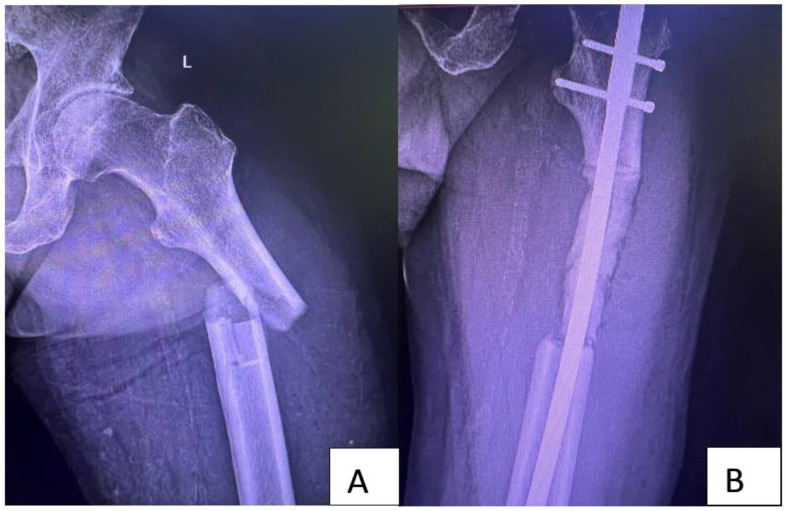
Preoperative X-ray (**A**) of a 56-year-old male with a history of recently diagnosed non-small-cell lung cancer (NSCLC) and a pathological bone fracture. After multidisciplinary evaluation, no other secondary tumor was found. Due to the fracture, the approach chosen was metastatic resection followed by intramedullary nail and acrylic bone cement (**B**). The conduct in the case of this patient was to carry out periodic imaging evaluations and to replace the nail if the local and general situation allowed a push-through modular component. Imaging evaluations performed at 12 months detected multiple metastases, so replacement with a modular component was abandoned.

**Table 1 medicina-58-01463-t001:** Systemic treatment options depending on genetic alterations.

Genetic Alteration	Possible Systemic Treatment
EGFR exon 19 deletion or L858R mutation	osimertinib
EGFR S768I, L861Q, G719X mutation	afatinib or osimertinib
EGFR insertion exon 20	amivantamab or mobocertinib
KRAS G12C mutation	sotorasib
ALK rearrangement	alectinib or lorlatinib or brigatinib
ROS1 rearrangement	entrectinib or crizotinib
NTRK 1/2/3 gene fusion	larotrectinib or entrectinib
MET exon14 skipping mutation	capmatinib or tepotinib
RET rearrangement	serpercatinib

## Data Availability

Further data concerning the study can be obtained by contacting the corresponding author.

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
