# Peer review of "Management of Lung Cancer Presenting with Solitary Bone Metastasis"

_medicina, 2022, doi:10.3390/medicina58101463_

Round 1

Reviewer 1 Report

Dear Editor and Authors,

Thank you for giving me the opportunity to review this work titled “Management of Lung Cancer Presenting with Solitary Bone Metastasis” by Dr. Claudiu Eduard Nistor and colleagues from the Department Thoracic Surgery at Dr Carol Davila University Hospital in Bucharest, Romania.

I must say that as a thoracic surgeon the notion that the authors present and review the data on is quite intriguing and it peaked my interest just from reading the title! As the authors note in their very well structured and presented introduction, the management of patients with lung cancer and a solitary bone metastasis is very sparse and mostly anecdotal/case reports. This is of course because in most cases there is multiple bone metastases in contrast to brain or adrenal which can be solitary and thus are unresectable.

This is overall a very nice review. I particularly enjoyed the recommendation regarding surgery for lung cancer with a solitary bone metastasis. They are concise and logical even though of course real data and guidelines are minimal/non-existent. Nevertheless, the authors have tried to put together some logical recommendations which I find acceptable.

Overall this a nice review, there isn’t as previously mentioned much about this entity in the literature and thus this small but concise review is of use to the scientific community putting together what is available in comprehensive way. The manuscript is well written and has some nice imaging included which break the monotony and add some visual stimulus to the reader.  Kind regards to all.

Author Response

Thank you for your kind review. The answer to the question of how these cases should be approached correctly can only be obtained after prospective multicenter studies that include a large number of patients. Maybe a pan-European study would be a solution.

Thank you

Reviewer 2 Report

Lung neoplasm is the main cause of cancer-related mortality, and bone metastasis is among the most common secondary tumors.

The vast majority of patients present with multiple bone metastases, and the prognosis of these patients is poor. However, a very small number, approximately 7%, of patients diagnosed with non-small-cell lung neoplasm present a single metastasis at the initial evaluation. The optimal approach for these patients is not fully established.

The authors reviewed 41 papers addressing this subject, analyzed a small category of patients with lung neoplasms with a single bone metastasis, suggested that synchronous surgical treatment led to significant increases in medium- and long-term survival.

The authors also suggested that sequencing of local versus systemic treatment should always be discussed within the multidisciplinary team. Targeted systemic therapies, radical surgical treatment along with local treatments will increase the survival of patients.

The present study is well organized, however, there is no available information on the pathology of those lung neoplasm, and this should be provided if available.

Author Response

Thank you very much for your kind review. The study is designed from the perspective of the treatment of bone metastases secondary to pulmonary neoplasm, be they single or multiple, and less from the perspective of non-metastatic pulmonary neoplasm. For the cases presented as examples, we do not have detailed information about the pathology of the lung neoplasm.

Thank you

Reviewer 3 Report

Good review article on management of lung cancer with bone metastasis.

Minor comments:

Please rephrase the lines 27-30, 47-49. They are exactly same as what you have in Abstract.

Author Response

Thank you very much for the review. We will rephrase the lines you mentioned. 

Thank you